# Fizzy-Related dictates A cell cycle switch during organ repair and tissue growth responses in the *Drosophila* hindgut

Erez Cohen[1], Scott R Allen[1], Jessica K Sawyer[2], Donald T Fox[1,2,3]*

[1]Department of Cell Biology, Duke University School of Medicine, Durham, United States; [2]Department of Pharmacology & Cancer Biology, Duke University School of Medicine, Durham, United States; [3]Regeneration Next Initiative, Duke University School of Medicine, Durham, United States

**Abstract** Ploidy-increasing cell cycles drive tissue growth in many developing organs. Such cycles, including endocycles, are increasingly appreciated to drive tissue growth following injury or activated growth signaling in mature organs. In these organs, the regulation and distinct roles of different cell cycles remains unclear. Here, we uncover a programmed switch between cell cycles in the *Drosophila* hindgut pylorus. Using an acute injury model, we identify mitosis as the response in larval pyloric cells, whereas endocycles occur in adult pyloric cells. By developing a novel genetic method, DEMISE (Dual-Expression-Method-for-Induced-Site-specific-Eradication), we show the cell cycle regulator Fizzy-related dictates the decision between mitosis and endocycles. After injury, both cycles accurately restore tissue mass and genome content. However, in response to sustained growth signaling, only endocycles preserve epithelial architecture. Our data reveal distinct cell cycle programming in response to similar stimuli in mature vs. developmental states and reveal a tissue-protective role of endocycles.
DOI: https://doi.org/10.7554/eLife.38327.001

*For correspondence:
don.fox@duke.edu

Competing interests: The authors declare that no competing interests exist.

## Introduction

Throughout development, cell cycle regulation is altered to build tissues and organs. Examples include the lack of gap phases to rapidly increase cell number in embryos of many species (*Edgar and O'Farrell, 1990*; *Graham and Morgan, 1966*; *McKnight and Miller, 1977*; *Newport and Kirschner, 1982*), ploidy-reducing meiotic divisions to produce haploid gametes (*Kleckner, 1996*), or ploidy-increasing cycles that enable rapid post-mitotic tissue growth (*Calvi, 2013*; *Fox and Duronio, 2013*; *Hua and Orr-Weaver, 2017*; *Edgar et al., 2014*).

After development is completed, a diversity of cell cycle regulation is also found in mature adult tissues during injury repair. In many stem cell-based tissues, or in highly regenerative organs/organisms, mitotic cell cycles restore pre-injury cell number and size (*Jiang et al., 2009*; *Mascré et al., 2012*; *Mauro, 1961*; *Poss et al., 2002*; *Ryoo et al., 2004*; *Yan et al., 2012*). In contrast, we and others previously defined injury responses in the adult *Drosophila* hindgut and abdomen, tissues that lack mitotic divisions (*Fox and Spradling, 2009*; *Losick et al., 2013*; *Sawyer et al., 2017*). In these adult tissues, injury leads to an increase in cellular ploidy through endocycles (G/S cycles without M phase, see cell cycle nomenclature section of Materials and methods). These *Drosophila* responses have clear parallels in the hypertrophic tissue injury repair of mammals. Injured mammalian hearts alter their cell cycle programming from mitotic to ploidy-increasing cell cycles during defined periods in development (*Porrello et al., 2011*). As a result, cardiac cells typically undergo hypertrophy instead of hyperplasia in response to injury or sustained tissue growth signals such as from the Ras/Raf pathway (*Hunter et al., 1995*; *Porrello et al., 2011*; *Wu et al., 2011*; *Yu et al.,*

**eLife digest** How does an injured organ replace cells that have died as a result of the damage? Making new cells might seem like the best option, because this would restore the organ to how it looked before the injury. To make new cells, existing cells in the organ divide. But not all organs make new cells to repair damage. Instead, some organs make their remaining cells bigger – a process known as hypertrophy – to fill the space created by the injury.

Cohen et al. have now developed a technique to investigate which method of repair a damaged organ uses. The technique uses genetic engineering to create an injury in a specific tissue in fruit flies, while also altering the activity of other genes that might affect how the tissue responds to the injury. Using the technique to study injuries to part of the gut that remains the same throughout a fly's life revealed that fly larvae repair this damage by creating new cells. However, adult flies repair the same injuries using hypertrophy.

Cohen et al. found that a gene known as 'fizzy-related' helps to control how the organ repairs damage. The fizzy-related gene produces a protein that stops cells dividing, which forces the cells to grow to repair any injuries to the organ. Adult flies that lacked the gene repaired their guts through cell division instead of by hypertrophy. This did not affect how well minor injuries to the gut were repaired. However, under conditions of more extreme tissue injury cell division distorted the gut and led to leakiness of gut contents.

Hypertrophy has been seen in injured human organs, including the heart, liver and kidneys. This was thought to be an abnormal response, but the results presented by Cohen et al. suggest that hypertrophy may instead help to protect the organs during repair. Further research into the role of hypertrophy could reveal ways to regenerate damaged organs, perhaps by targeting the activity of the fizzy-related gene.

DOI: https://doi.org/10.7554/eLife.38327.002

*2015*). In the liver, injury can cause either mitotic or ploidy-increasing cell cycle responses (*Gentric et al., 2015*; *Miyaoka et al., 2012*; *Nagy et al., 2001*). Recently, the mouse kidney was discovered to endocycle in response to acute injury (*Lazzeri et al., 2018*). Thus, both during development and in post-development injury contexts, diverse cell cycle responses can occur.

Little is known about the molecular programming or functional consequence of distinct cell cycles used in injured adult tissues. One technical limitation to studying this question is the ability to conduct carefully targeted injury experiments while simultaneously performing genetic studies. Here, we introduce a new system termed Dual-Expression-Method-for-Induced-Site-specific-Eradication (DEMISE), which enables us to finely control and independently manipulate both injury and genetics in our system. Using this system, we uncover developmental regulation and functional differences between two injury-induced cell cycle programs in the *Drosophila* hindgut pyloric epithelium.

The pyloric epithelium is the only segment of the hindgut to persist throughout the lifespan of the fly. Taking advantage of this persistence, we uncover that when injured the same way, pyloric cells undergo mitotic cycles in larvae but undergo endocycles in mature adults. Further, using this tissue model and our new genetic system, we demonstrate that active inhibition of mitotic cyclins by the conserved Anaphase Promoting Complex/Cyclosome (APC/C) regulator Fizzy-related (Fzr) underlies the alteration in injury-induced cell cycle programs in the pyloric epithelium. We identify that by blocking entry into mitosis, Fzr-mediated endocycles protect the adult pylorus against disruptions in epithelial architecture and permeability under conditions of sustained tissue growth signaling. Together, our results suggest that in some mature tissues, endocycles may represent a tradeoff between loss of regenerative capacity and preservation of tissue architecture.

## Results

### Drosophila hindgut pyloric cells accurately replace lost genome content using two developmentally distinct responses

We previously demonstrated that the adult *Drosophila* hindgut pyloric epithelium (hereafter- pyloric cells) provides an accessible model to study tissue injury repair through endocycles (*Fox and*

*Spradling, 2009*; *Losick et al., 2013*; *Sawyer et al., 2017*). Unlike many adult intestinal cells, pyloric cells are also a constituent segment of the larval hindgut. During metamorphosis, pyloric cells act as facultative progenitor cells, as they remodel the hindgut by undergoing mitotic cell division to both expand the larval pylorus into its adult form while also producing cells of the adult ileum, which replace the histolysed larval ileum (*Figure 1A*, *Fox and Spradling, 2009*; *Robertson, 1936*; *Sawyer et al., 2017*; *Takashima et al., 2008*). Thus, pyloric cells are capable of distinct cell cycles-mitotic cycles during organ remodeling (at metamorphosis) and endocycles during tissue injury repair (at adulthood).

We tested two possible models for the difference in pyloric cell cycle programs. In one model, pyloric cell cycle program is dictated by the stimulus: that is, induced apoptotic injury promotes endocycles while developmental gut histolysis promotes cell division. In a second alternative model, developmental status of the pylorus may solely govern cell cycle status, regardless of the type of injury. To distinguish between these possibilities, we injured the larval pylorus at the last developmental stage before metamorphosis (wandering third larval instar, L3) and allowed the animals to progress to adulthood. For tissue injury, we used temporal and spatial control of the pro-apoptotic genes *head involution defective* (*hid*) and *reaper* (*rpr*), as before (Materials and methods, *Figure 1B*, '1', *Fox and Spradling, 2009*; *Losick et al., 2013*; *Sawyer et al., 2017*). As a comparison, we injured adult pyloric cells using the same scheme (*Figure 1B*, '2'). In both cases, we confirmed that our injury protocol causes pyloric cell death (*Figure 1—figure supplement 1A–E*). To clearly demarcate the recovered pylorus, we used reporters of pyloric boundaries (*Figure 1C–E*). We then compared adult flies recovered from either larval or adult injury to identify any differences in the mode of pyloric recovery.

While animals recovered from both larval and adult injury show no obvious defects in recovery of pyloric tissue mass, the response to larval and adult injury is strikingly different. When compared against uninjured animals (*Figure 1C–C'*), adult pyloric cells recovered from a larval injury (*Figure 1D–D'*) show no change in cell ploidy and remain diploid (*Figure 1G*). Additionally, larval injury does not change the number or size of cells recovered in adults (*Figure 1H*, *Figure 1—figure supplement 1F–F',H*). The ability to produce an adult gut of normal cell number and ploidy was impressive given that we eliminated a high percentage of larval pyloric cells by injury (*Figure 1—figure supplement 1E*). In contrast, comparable injury to the adult pylorus persistently increases ploidy, decreases cell number, and increases cell size, as we previously reported (*Figure 1E,G,H*, Losick et al., 2013). However, it remained possible that the number of surviving cells following tissue injury dictates the pyloric response. To test this idea, we took advantage of the ability to finely tune injury level in our system (Materials and methods) and used varying durations (12–48 hr) of apoptotic gene expression to produce a linear decrease in cell survival (*Figure 1F,H*). Regardless of the severity of adult injury, cell number does not recover, whereas cell ploidy and cell size increases (*Figure 1F,G*, *Figure 1—figure supplement 1G–I*). Thus, developmental stage and not injury severity dictates the pyloric cell cycle response.

Our quantitation of adult pyloric ploidy after injury revealed a proportional tissue injury response-ploidy increases closely track with the degree of cell number loss (*Figure 1G–H*, $R^2 = 0.97$, p<0.002). These results suggested that regardless of developmental stage, the pylorus remains capable of closely replacing the number of genomes lost to injury. To test this model, we quantified the total genome content per pylorus (number of cells x ploidy) following injury recovery (Materials and methods). Our data show that regardless of injury severity or developmental stage, the pre-injury pyloric total genome content is fully recoverable (*Figure 1I*). These results indicate that through developmentally distinct cell cycle injury responses, the pylorus accurately restores genomic content in proportion to the injury stimulus (*Figure 1J*).

## Distinct cell cycle programs underlie the distinct pyloric injury responses

We next analyzed the cell cycle responses of the two distinct pyloric injury recovery modes. Our cell ploidy and cell number quantitation suggest that larval pyloric cells are capable of producing new cells to regenerate the injured gut during metamorphosis, whereas adult pyloric cells have lost this regenerative ability. To assess this, we first traced the lineage of uninjured and injured larval hindgut cells during metamorphosis (*Figure 2A*, *Figure 2—figure supplement 1A*). We previously used a low-background, low-frequency clonal marking system to demonstrate that single larval pyloric cells

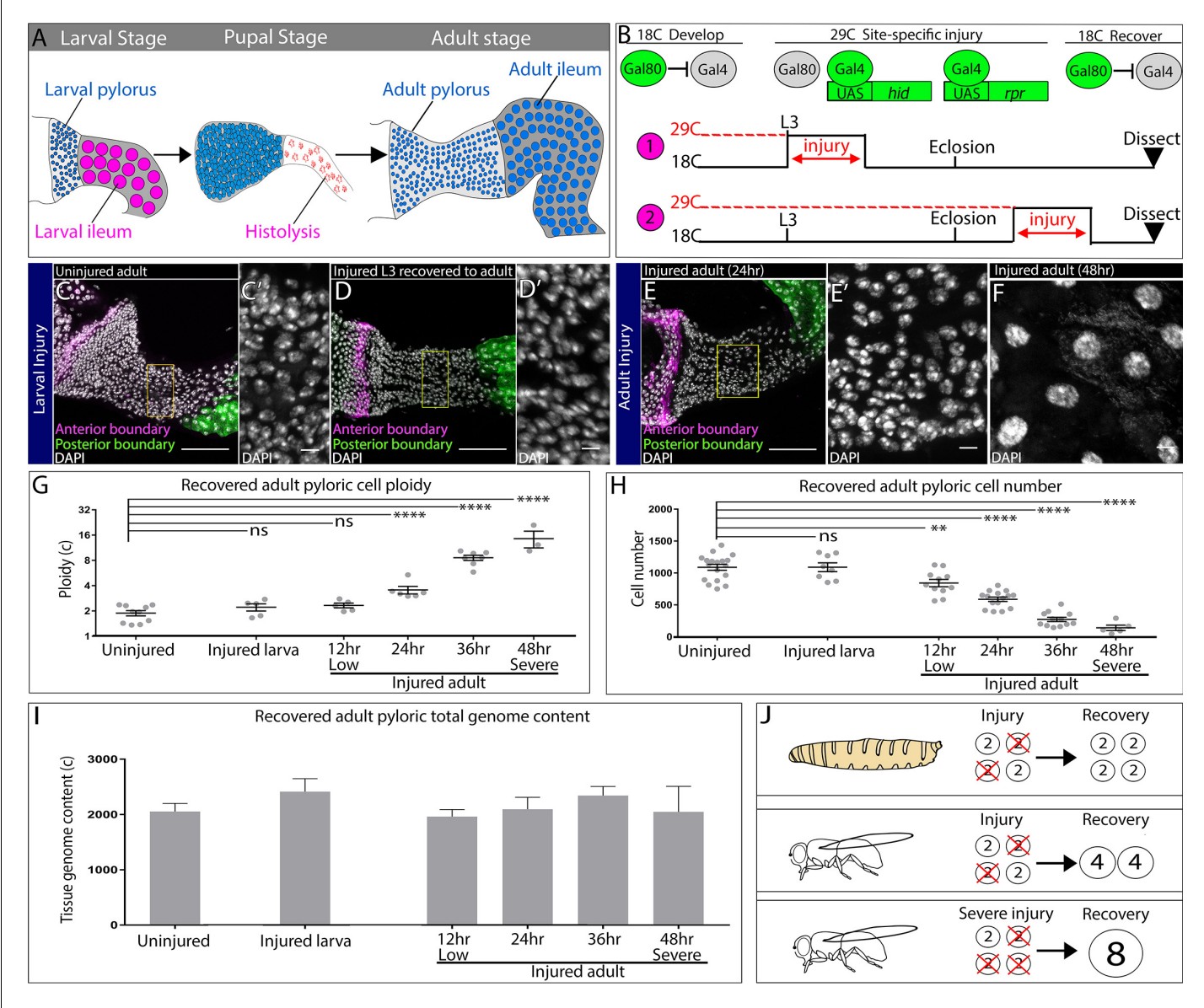

**Figure 1.** Injured hindgut pyloric cells replace lost genome content using two distinct responses. (**A**) Schematic of pyloric development. (**B**) Experimental injury scheme (see Results and Materials and methods). Numbers 1 and 2 are referenced in the text. (**C–F**) Adult pylori. Anterior boundary marked by *wg-LacZ* (magenta), posterior boundary marked by Vha16-GFP (green), and nuclei (DAPI, white). Yellow box highlights the region shown in the adjacent high magnification inset (**C',D',E'**). (**C–C'**) Uninjured adult pylorus. (**D–D'**) Injured L3 recovered to adult (**E–F**). Adult pylorus injured for 24 hr (**E–E'**) or 48 hr (**F**) and recovered for 5 days. (**G–H**) Quantification of pyloric ploidy (**G**) and cell number (**H**). (**I**) Quantification of pyloric total genome content. (**J**) Model of larval vs adult recovery from injury. For panels (**G–I**) data represent mean ± SEM, N ≥ 5 animals, at least two replicates. ANOVA, Dunnett's multiple comparisons. Scale bars (**C,D,E**) 50 µm, (**C',D',E',F'**) 5 um.

DOI: https://doi.org/10.7554/eLife.38327.003

The following figure supplement is available for figure 1:

**Figure supplement 1.** Supporting data related to *Figure 1*.

DOI: https://doi.org/10.7554/eLife.38327.004

produce, on average, either ~5 adult pyloric cells or two adult ileal cells during metamorphosis, and afterwards cease to divide (*Fox and Spradling, 2009*). We reproduced these data in uninjured animals (Materials and methods, *Figure 2C*, *Figure 2—figure supplement 1B–E*). By comparison, in adults recovered from larval injury, clone size increases by ~3 fold on average (*Figure 2D–D', G,H*, uninjured: 5.389 ± 0.5725 SEM, injured: 16.54 ± 1.863 SEM, p<0.0001). Our clonal data coupled with

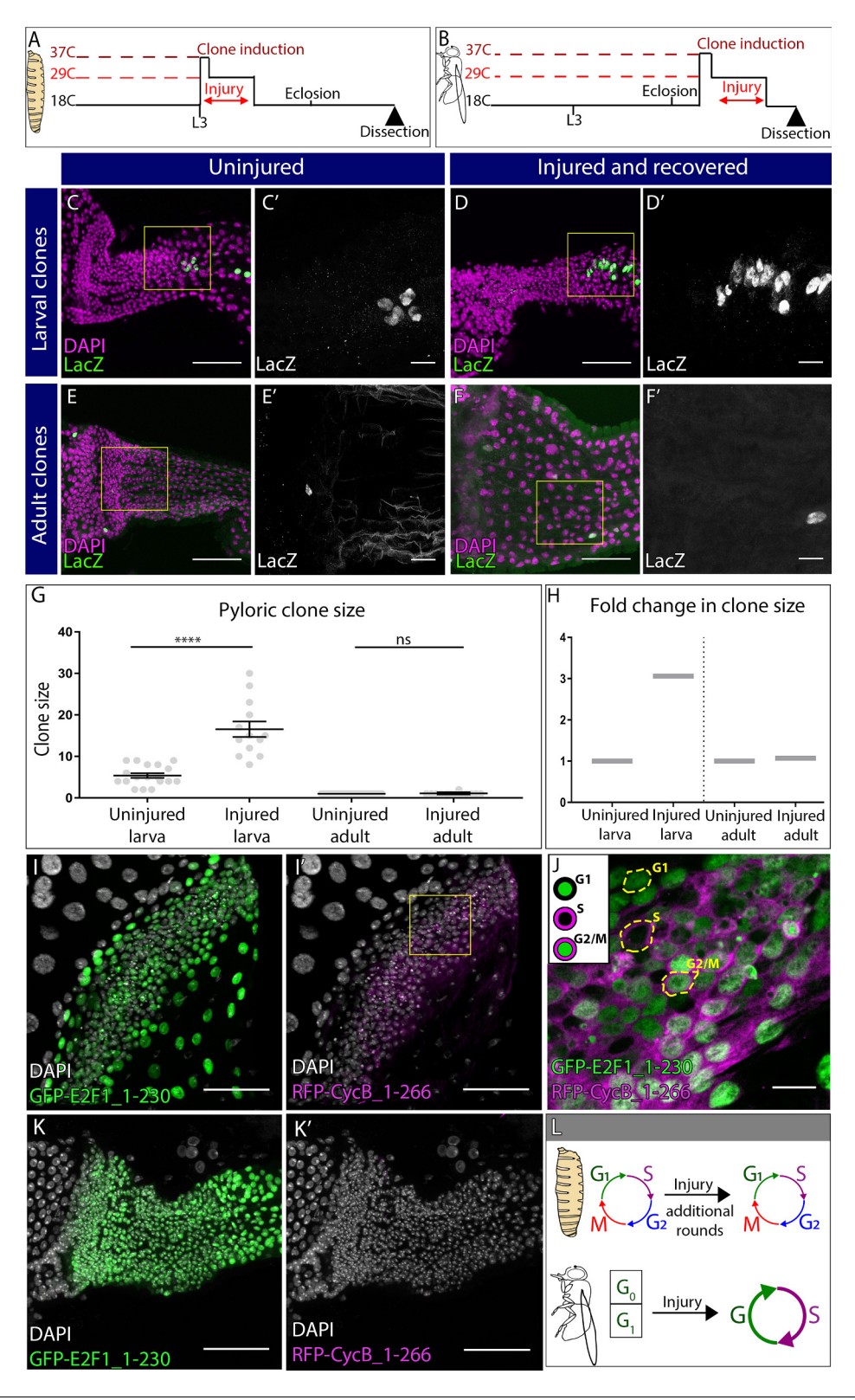

**Figure 2.** Distinct cell cycle programs underlie the distinct pyloric injury responses. (**A–B**) Schematic of clone and injury induction in larvae (**A**) and adults (**B**). (**C–F'**) Clones in adult pylori. Clones (*LacZ*, green) and nuclei (DAPI, magenta). Yellow box highlights the region shown in the adjacent high magnification inset (**C',D',E', F'**). (**C–D'**) Clones induced at L3 stage without injury (**C–C'**) or with injury (**D–D'**). (**E–F'**) Clones induced at adult stage without injury (**E–E'**) or with injury (**F–F'**). (**G**) Quantification of pyloric clone size ±injury. Data represent mean ± SEM, N ≥ 13 clones per condition,≥13
*Figure 2 continued on next page*

Figure 2 continued

animals per condition, at least two replicates. Unpaired two-tailed t-tests. (H) Fold change in clone size. (I–K') Fly-FUCCI animals expressing GFP-E2F1_1–230 (green), RFP-CycB_1–266 (magenta) and nuclei (DAPI, white). (I–I') Fly-FUCCI in the larval pylorus. (J) Cell cycle stages in larval pyloric cells. Yellow hash marks cell outline. (K–K') Fly-FUCCI in the adult pylorus. (L) Model of cell cycle utilized by larvae vs adults after injury. Scale bars (C,D,E,F,I, K) 50 µm, (C',D',E',F',J) =10 µm.

DOI: https://doi.org/10.7554/eLife.38327.005

The following figure supplement is available for figure 2:

**Figure supplement 1.** Supporting data related to *Figure 2*.

DOI: https://doi.org/10.7554/eLife.38327.006

our cell counts suggest that approximately 75% of larval cells were eliminated using our injury protocol and were then recovered by compensatory proliferation. Clone size in the ileum also increases in response to injury, consistent with the model that these cells derive from the larval pylorus (*Figure 2—figure supplement 1C–E*). Further, these data show that remaining larval pyloric cells (approximately 25%) have the capability of increasing their mitotic capacity to completely regenerate the adult hindgut pylorus and ileum following acute injury.

In contrast to larvae, inducing clones in adult pylori (*Figure 2B*) produces only single labeled cells (27/27 clones, *Figure 2E*). Our marking system does not require a cell division to generate single-labelled cells (*Figure 2—figure supplement 1A*). Using this system, we did not see any expansion of single labeled cells into mitotic clones even after 20–30 days of recovery from a severe injury (*Figure 2F–H*, uninjured: $1.00 \pm 0$ SEM, injured: $1.07 \pm 0.07$ SEM, p=0.99 and *Figure 2—figure supplement 1F–H*). In further support of our lineage data, the M-phase marker Phospho-Histone H3 (PH3) does not label adult pyloric cells (data not shown, *Fox and Spradling, 2009*; *Sawyer et al., 2017*), while it frequently labels larval pyloric cells (*Figure 2—figure supplement 1I*). However, both larvae and adult pyloric cells incorporate the S-phase marker EdU following injury (*Figure 2—figure supplement 1J,K*). Taken together, our cell cycle marker, lineage, and cell ploidy/number quantitation show that only larval pyloric cells divide while adult pyloric cells instead endocycle after injury.

Having determined the two distinct post-injury pyloric cell cycle responses, we next analyzed the pre-injury cell cycle status of both larval and adult pyloric cells. For this analysis, we used the Fly Fluorescent Ubiquitin-based Cell Cycle Indicator (Fly-FUCCI) system (*Zielke et al., 2014*). In the pylorus of uninjured larvae, the FUCCI components E2F1$_{1-230}$-GFP and Cyclin B$_{1-266}$-mRFP are expressed in patterns consistent with the presence of G1, S, and G2/M cells (*Figure 2I–J*). Coupled with our M phase data (*Figure 2—figure supplement 1I*), we find that both before and after injury, larval pyloric cells undergo mitotic cycles, although injury leads to additional rounds of mitotic cycles (*Figure 2L*). In contrast, 100% of uninjured adult pyloric cells are E2F1$_{1-230}$-GFP positive and Cyclin B$_{1-266}$-mRFP negative (*Figure 2K*), indicative of a quiescent/arrested G0/G1 state of these cells prior to their entry into injury-activated endocycles. Thus, pyloric injury at distinct developmental stages induces distinct cell cycle changes (*Figure 2L*).

## DEMISE, a new method for dual control of site specific genetic ablation and transgene expression

Having identified developmental stage as a determinant of the distinct pyloric cell cycle injury responses, we next sought to identify molecular regulation of these responses. Our apoptotic injury system takes advantage of the temperature sensitive Gal80$^{ts}$ repressor to control Gal4-mediated expression of apoptotic genes (*Brand and Perrimon, 1993*; *McGuire et al., 2004*). While this enables fine control over the developmental stage, tissue location, and degree of injury, it also prevents use of the numerous Gal4-mediated gene knockdown tools such as the large collection of transgenic *UAS-RNAi* and now CRISPR lines (*Dietzl et al., 2007*; *Lin et al., 2015*; *Ren et al., 2013*; *Wu et al., 2011*). This is because any transgene-expressing cell would be dead using our standard apoptotic method. An ideal system would instead allow for separate control over injury and transgene expression.

We thus developed such a system, which we term DEMISE (Dual-Expression-Method-for-Induced-Site-specific-Eradication). DEMISE combines the Gal4/Gal80$^{ts}$ and FLP/FRT systems to induce developmentally-timed, tunable tissue-specific injury in a mosaic fashion while independently expressing any *UAS*-driven transgene of interest in the same tissue. Compared to our previous injury system,

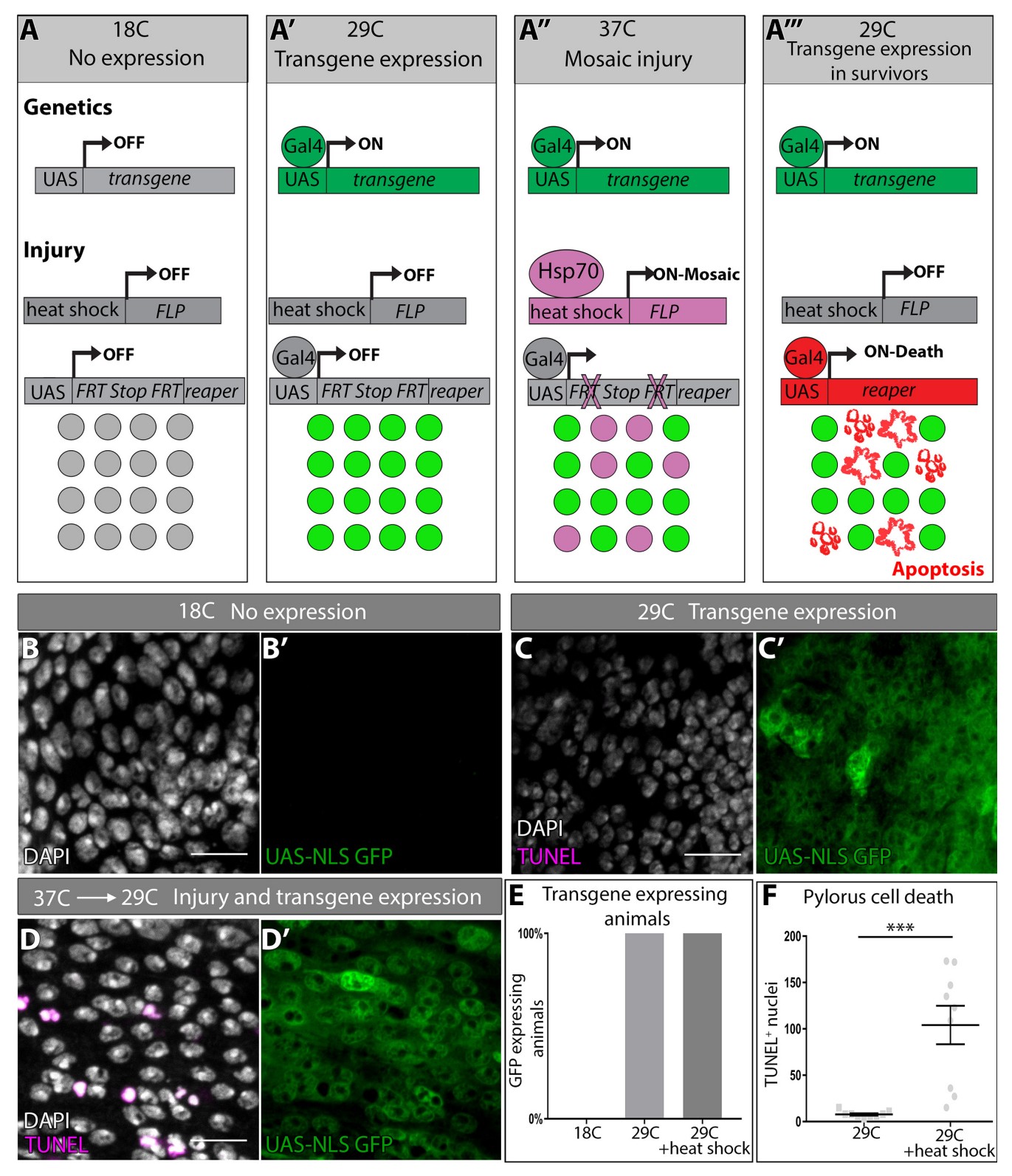

**Figure 3.** DEMISE, a novel tool for dual control of site-specific genetic ablation and transgene expression. (**A–A′′′**) DEMISE method and components (see Results and Materials and methods). (**B–D**) Adult pylori. Transgene expression marked by *UAS-NLS GFP* (green), cell death (TUNEL, magenta), and nuclei (DAPI, white). (**B–B′**) No transgene expression at 18C. (**C–C′**) Transgene expression and no injury at 29C. (**D–D′**) Transgene expression and injury

*Figure 3 continued*

following heat shock (37C). (**E**) Quantification of pyloric transgene expression at different DEMISE stages. N ≥ 5 animals. (**F**) Quantification of pyloric cell death following heat shock. Data represent mean ± SEM, N ≥ 5 animals, at least two replicates. Unpaired two-tailed t-test. Scale bars (**B–D'**) 10 µm.

DOI: https://doi.org/10.7554/eLife.38327.007

The following figure supplement is available for figure 3:

**Figure supplement 1.** Supporting data related to *Figure 3*.

DOI: https://doi.org/10.7554/eLife.38327.008

DEMISE adds an FRT-Stop-FRT cassette between a *UAS* promoter and the apoptotic *rpr* gene (*Figure 3A*). Thus, cell death and transgene expression are still under control of a tissue-specific Gal4 and temperature inactivation of Gal80$^{ts}$ (*Figure 3A'*) but cell death also requires FLP-mediated removal of the Stop cassette. Temporal control of injury, as well as injury strength, is then achieved by timing expression of FLP under heat shock (hs) control (*Figure 3A''*). In parallel, Gal4/Gal80$^{ts}$ drives expression of a transgene of interest independent of FLP-mediated cell death (*Figure 3A', A''*). The result is that all cells of a given tissue can express a transgene (such as a gene knockdown RNAi), while a fraction of these cells (determined by degree of FLP induction) will be eliminated by injury (*Figure 3A'''*). This experimental design enables one to use the vast *UAS*-driven *Drosophila* transgene collection while also inducing a precise injury.

To test the efficacy of DEMISE, we generated a series of *UAS-FRT-Stop-FRT-reaper* insertions in the *Drosophila* genome (Materials and methods). We identified FLP/DEMISE combinations with both low background cell death (lack of leaky Stop cassette excision) along with robust apoptosis induction upon heat shock (*Table 1*). We began by examining larval wing imaginal discs, a frequently used model to study tissue regeneration in response to apoptotic injury (*Halme et al., 2010*; *Harris et al., 2016*; *Smith-Bolton et al., 2009*). We drove DEMISE expression in the posterior wing disc compartment with *engrailed (en) > Gal4*. Using temperature shifts and a *UAS-GFP* reporter, transgene expression is induced in the expected pattern (*Figure 3—figure supplement 1A–C*). Our transgene expression conditions do not induce cell death, as assayed by TUNEL (*Figure 3—figure supplement 1B*). However, upon heat shock to induce FLP expression, robust cell death occurs in a mosaic pattern specifically in the *engrailed* domain (*Figure 3—figure supplement 1C–D*, Materials and methods).

We then returned to the hindgut system and performed similar transgene expression controls (*Figure 3B–D*). Again, using our most efficacious DEMISE/FLP combination in the hindgut, we can achieve a low background level of cell death (*Figure 3C,D,F*), followed by robust induction of cell death upon heat shock that occurs independently of transgene expression (*Figure 3C–F*, Materials and methods). Our results establish DEMISE as a new method for inducing temporal and site-specific injury while maintaining independent control of transgene expression.

**Table 1.** DEMISE combinations tested in the hindgut

| Flip-out reaper insertion | Genotype | [ry[+t7.2]=hsFLP]1 | [ry[+t7.2]=hsFLP]12 |
|---|---|---|---|
| *UAS*-rprFLPout 10–3 | *pUAST-FRT-STOP-FRT-rpr*/CyO #Insertion 10–3 | Leaky | Not leaky (used in paper) |
| *UAS*-rprFLPout 9–1 | *pUAST-FRT-STOP-FRT-rpr*/TM3*Sb* #Insertion 9–1 | Leaky | Not leaky |
| *UAS*-rprFLPout 5–1 | *pUAST-FRT-STOP-FRT-rpr*/TM3*Sb* #Insertion 5–1 | Leaky | |
| *UAS*-rprFLPout 4–1 | *pUAST-FRT-STOP-FRT-rpr*/TM3*Sb* #Insertion 4–1 | Not leaky | |
| *UAS*-rprFLPout 10–1 | *pUAST-FRT-STOP-FRT-rpr*/CyO #Insertion 10–1 | Leaky | Leaky |
| *UAS*-rprFLPout 10–2 | *pUAST-FRT-STOP-FRT-rpr*/TM3Sb #Insertion 10–2 | Leaky | Leaky |

DOI: https://doi.org/10.7554/eLife.38327.009

# DEMISE reveals *fizzy-related* as a regulator of injury-mediated cell cycle programming

After establishing DEMISE as a novel system to study molecular regulators of injury responses, we sought to identify genes that dictate which pyloric cell cycle program is induced by injury. Our pre-injury FUCCI data (*Figure 2I–K*) provided a potential clue, as larval but not adult uninjured pyloric cells co-express the GFP-E2F1$_{1-230}$ and RFP-CycB$_{1-266}$ FUCCI reporters, indicative of a G2/M state (*Zielke et al., 2014*). This led us to hypothesize that inhibition of mitotic cyclins in the adult pylorus may prevent entry into mitotic cycles after injury. Fizzy-related (Fzr), the *Drosophila* homolog of mammalian FZR1/CDH1, is a binding partner of the APC/C, which facilitates degradation of mitotic cyclins including Cyclin B. Fzr is also a well-known regulator of the endocycle in many cell types (*Lehner and O'Farrell, 1989*; *Schaeffer et al., 2004*; *Sigrist and Lehner, 1997*; *Stormo and Fox, 2016*) and *fzr* mutant cells can ectopically undergo mitotic cycles instead of endocycles (*Schaeffer et al., 2004*; *Schoenfelder et al., 2014*; *Sigrist and Lehner, 1997*). Thus, increased *fzr* expression in the adult pylorus is a plausible mechanism for the altered injury responses we observe.

We first asked whether *fzr* is upregulated in the adult pylorus. Indeed, using two independent *fzr* enhancer traps, *fzr* expression is high in the adult pylorus, whereas *fzr* is undetectable in the larval pylorus (*Figure 4A–B'*, *Figure 4—figure supplement 1L–M'*). We next asked if elimination of *fzr* is sufficient to revert the adult pyloric injury response (endocycles) to the larval injury response (mitotic cycles). Using DEMISE, we expressed *fzr RNAi* in the hindgut throughout development (*Figure 4C*). Without injury, *fzr* knockdown did not noticeably alter hindgut development with the exception of the rectal papillae, where we previously identified a role for *fzr* in the pre-mitotic endocycles (*Schoenfelder et al., 2014*, *Figure 4—figure supplement 1A–B*). Following apoptotic injury induction, both control and *fzr RNAi*-expressing animals contain both TUNEL positive nuclei and pycnotic nuclei (*Figure 4D–F*), which are more prevalent at the anterior pylorus as previously described (*Fox and Spradling, 2009*; *Sawyer et al., 2017*). Further, lack of *fzr* does not alter the number of adult pyloric cells entering the cell cycle as seen by BrdU staining (*Figure 4—figure supplement 1C*).

Strikingly, following 5 days of recovery from adult injury, the adult pylorus of *fzr* flies completely restores pre-injury cell number and ploidy (*Figure 4G–J*). Further, during injury recovery, mitotic cells are visible in the adult *fzr* pylorus (5/7 animals), but not in control animals (0/17 animals, as assayed by PH3 and the centrosomal marker Centrosomin- Cnn, *Figure 4K–L'''*). We noted one additional cell cycle alteration in *fzr* animals. Endocycles frequently exhibit under-replication of late-replicating sequences, which in *Drosophila* cluster in a DAPI 'bright-spot' (*Belyaeva et al., 1998*; *Endow and Gall, 1975*; *Fox and Duronio, 2013*; *Gall et al., 1971*; *Nordman et al., 2011*; *Schoenfelder and Fox, 2015*; *Edgar et al., 2014*). In control adults, pyloric cells undergoing injury-induced endocycles only exhibit early replication patterns (as assayed by BrdU, *Figure 4—figure supplement 1D*), consistent with the S-phase pattern of many endocycling cells that undergo under-replication. In agreement with this idea, DNA FISH shows that a satellite DNA repeat that is commonly under-replicated in endocycling cells (*Endow and Gall, 1975*) does not increase in intensity in proportion to the obvious nuclear size increase induced by adult pyloric injury (*Figure 4—figure supplement 1N–O*). In contrast, *fzr* animals instead exhibit both early and late-S patterns after injury (*Figure 4—figure supplement 1E*). This pattern suggests that *fzr* loss may also enable progression through late S-phase, possibly due to a role for the *fzr* target cyclin A in promoting late replication (*SalleSallé et al., 2012*). Taken together, we find the two distinct injury-induced cell cycle programs in the pylorus are dictated by the action of Fzr, a conserved negative regulator of mitotic cyclins (*Figure 4M*).

Previous work on developmentally programmed switches from mitotic cycles to endocycles in *Drosophila* follicle cells also implicated a role for Fzr (*Schaeffer et al., 2004*). In this context, *fzr* is regulated upstream by Notch signaling and its transcriptional target Hindsight. Further, Notch frequently promotes endocycles in development and in adult tissue homeostasis (*Deng et al., 2001*; *Guo and Ohlstein, 2015*; *Von Stetina et al., 2018*). Therefore, we sought to determine if Notch is also used in an acute injury context to regulate the pyloric cell cycle program. Either with or without injury, adult pyloric cells do not express Hindsight, suggesting that this Notch target is not involved in the pyloric cell cycle injury response (*Figure 4—figure supplement 1F–G*). Further, expressing dominant negative *Notch* receptor throughout pupation using DEMISE does not prevent injury-mediated endocycles in the adult pylorus (*Figure 4—figure supplement 1H–I*), despite the fact that we observe expected *Notch* phenotypes in the rectal papillae (*Figure 4—figure supplement 1J–K*,

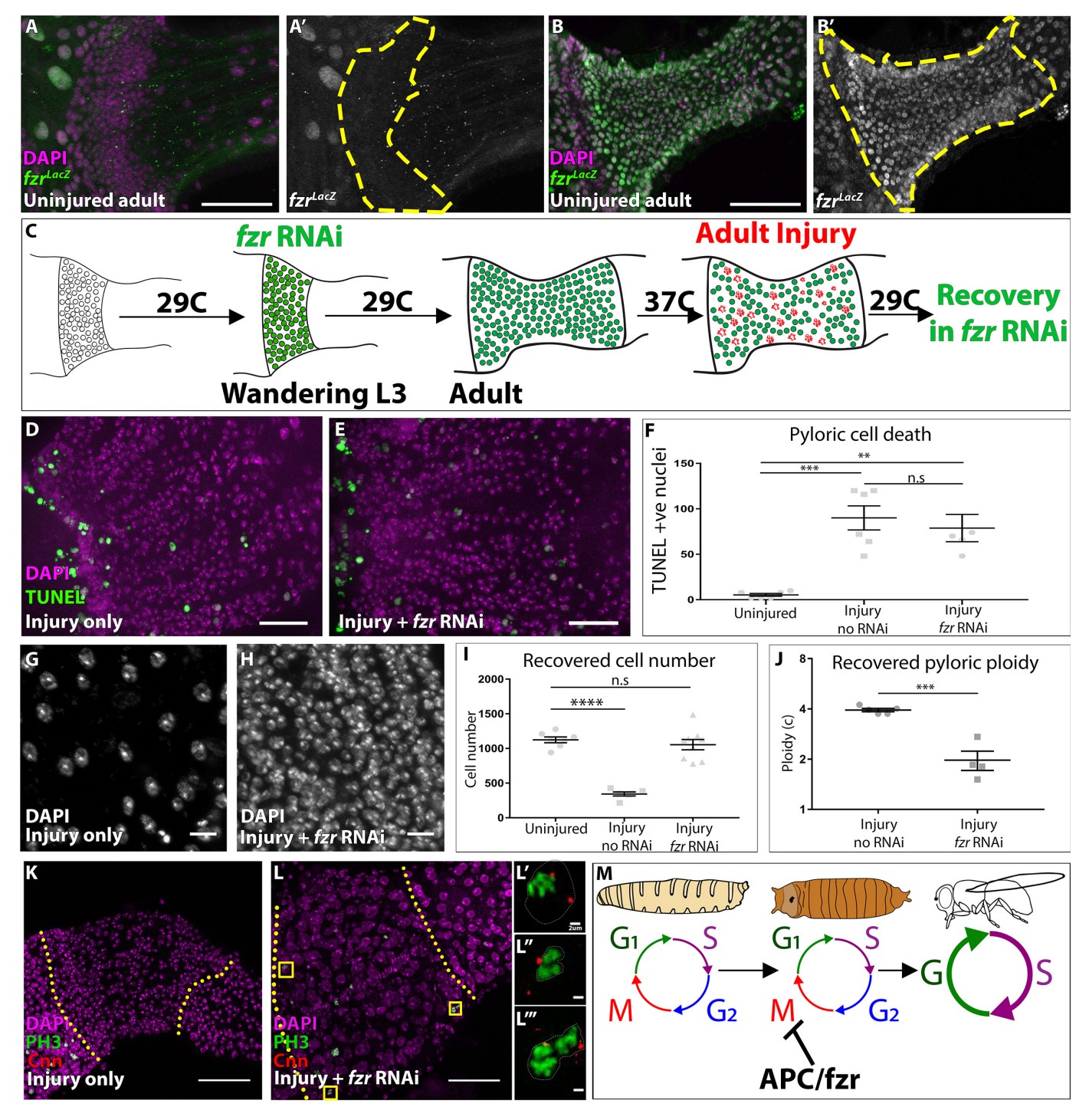

**Figure 4.** *fizzy-related* is a regulator of injury-mediated cell cycle programming. (**A–B'**) Expression of the *fzr^G0418* enhancer trap (*LacZ* expression, green or grey), nuclei (DAPI, magenta) in uninjured larval pylorus (**A–A'**) or adult pylorus (**B–B'**). Yellow dashed line marks pyloric boundaries. (**C**) Schematic of DEMISE, apoptosis (red) and *fzr RNAi* expression (green). (**D–E**) Adult pylori. Cell death (TUNEL, green) and nuclei (DAPI, magenta). (**D**) Injury only (**E**) injury +*fzr* RNAi. (**F**) Quantification of pyloric cell death ±*fzr* RNAi. Data represent mean ± SEM, N ≥ 5 animals, at least two replicates. ANOVA with Tukey's multiple comparisons. (**G–H**) Pyloric nuclei (DAPI, white) after injury and 5 days recovery. (**G**) injury only. (**H**) Injury + *fzr* RNAi. (**I**) Quantification of pyloric cell number after 5 days recovery. Data represent mean ± SEM, N ≥ 5 animals, at least two replicates per condition. ANOVA with Dunnett's multiple comparisons. (**J**) Quantification of pyloric ploidy after 5 days recovery. Data represent mean ± SEM, N ≥ 4 animals, at least two replicates. Unpaired two-tailed t-test. (**K–L'''**) Injured adult pylori. PH3 (green), Centrosomin (red), and nuclei (DAPI, magenta). Dashed line indicates pyloric

*Figure 4 continued on next page*

*Figure 4 continued*

boundaries. (**K**) Injury only (**L**) Injury + *fzr* RNAi. Yellow boxes highlight the region shown in adjacent high magnification insets (**L'–L''**). (**M**) Proposed model for the role of APC/Fzr(Cdh1) in regulation of injury cell cycle responses. Scale bars (**D,E**) 20 μm, (**G,H**) 5 μm, (**K,L**) 50 μm (**L'–L'''**) 2 μm.

DOI: https://doi.org/10.7554/eLife.38327.010

The following figure supplement is available for figure 4:

**Figure supplement 1.** Supporting data related to *Figure 4*.

DOI: https://doi.org/10.7554/eLife.38327.011

*Fox et al., 2010*; *Schoenfelder et al., 2014*). Thus, unlike previous studies on developmental endocycles, our results suggest a *Notch*-independent, *fzr*-dependent alteration in injury cell cycle responses in the *Drosophila* pylorus.

## Endocycles preserve intestinal architecture in response to sustained tissue growth signaling

Our studies of *fzr* animals showed that the normally diploid adult pylorus is primed to endocycle rather than maintain cellular ploidy under injury conditions. Both mitotic cycles and endocycles are capable of restoring pre-injury genome number throughout the tissue. This raises the question of whether there are any benefits to the switch away from mitosis and towards endocycles following cell cycle re-entry in the adult pylorus. One possibility that we considered is that, in response to pro-growth signaling, endocycles may be better at preserving tissue architecture than mitotic cycles. Activation of the pro-growth signaling Ras pathway, which is well-known to drive cell number increases in many contexts, is also linked to ploidy increases in cardiac tissue in flies and mammals (*Hunter et al., 1995*; *Wu et al., 2011*; *Yu et al., 2015*). We thus examined if constitutive activation of the small GTPase Ras ($Ras^{V12}$) could be used as an experimental tool to ask whether a tissue undergoing endocycles or mitotic cycles responds differently under conditions of sustained tissue growth signaling.

Using similar methods to our injury induction protocol (*Figure 5A*), we expressed $Ras^{V12}$ in larvae and then examined the resulting adults (*Figure 5B*, '1'). The pylorus of these animals is extremely expanded in size (*Figure 5C,D*). Cell number counts and size measurements show that larval $Ras^{V12}$ expression nearly triples the adult pyloric cell number without a substantial change in nuclear size (*Figure 5G,H*). This larval $Ras^{V12}$ response parallels the mitotic cell cycle response to injury in larvae, with the exception of a tissue overgrowth phenotype, which is likely due to sustained $Ras^{V12}$ expression overriding normal growth suppression signals. We next drove $Ras^{V12}$ expression in the adult (*Figure 5B*, '2'), again in the absence of injury. As with larval $Ras^{V12}$ expression, the adult pylorus expands in size following $Ras^{V12}$ expression (*Figure 5C,E*). However, and in parallel with our injury results, we observe no increase in cell number, while instead the nuclear size increases (*Figure 5G, H*). These data are consistent with $Ras^{V12}$ inducing endocycles in the mature adult pylorus. Thus, sustained growth signaling through $Ras^{V12}$ expression can be used to mimic both the larval and adult injury cell cycle responses found in the pylorus.

We next used $Ras^{V12}$ expression as a model to examine whether endocycles or mitotic cycles confer any tissue-level difference in the adult pylorus under conditions of prolonged tissue growth signaling. To ask this question, we compared $Ras^{V12}$ expressing adults to adults expressing both $Ras^{V12}$ and *fzr* RNAi (*Figure 5B* '3'). As in our injury studies, *fzr* suppresses nuclear size expansion in $Ras^{V12}$ expressing adults (*Figure 5C,F,H*). These results are consistent with a requirement of *fzr* for $Ras^{V12}$-driven endocycles in this tissue. Further, the overall pyloric cell number increases ~20% above controls in *fzr* RNAi + $RasV^{V12}$ animals, consistent with a role of *fzr* in preventing aberrant cellular hyperplasia in the adult pylorus (*Figure 5C,E–G*).

In further examination of $Ras^{V12}$ vs. *fzr* RNAi +$RasV^{V12}$ animals, we noticed an important difference with regards to intestinal epithelial architecture. Normally, the pyloric epithelia consists of a single cylindrical layer surrounding the hindgut lumen (*Figure 5I–I''*, *Figure 5—figure supplement 1A–A'*). The tissue expansion caused by $Ras^{V12}$ expression does not substantially distort this epithelial architecture (*Figure 5J–J'*). In contrast, in *fzr* RNAi +$RasV^{V12}$ animals, we noticed severe recurring abnormalities. Specifically, 23.3% (±3.3% SEM) of *fzr* RNAi +$RasV^{V12}$ animals examined show severe distortions of the intestinal epithelial architecture, either into or away from the intestinal lumen (*Figure 5K–K'*, *L*, *Figure 5—figure supplement 1B–C'''*). These distortions were not caused by

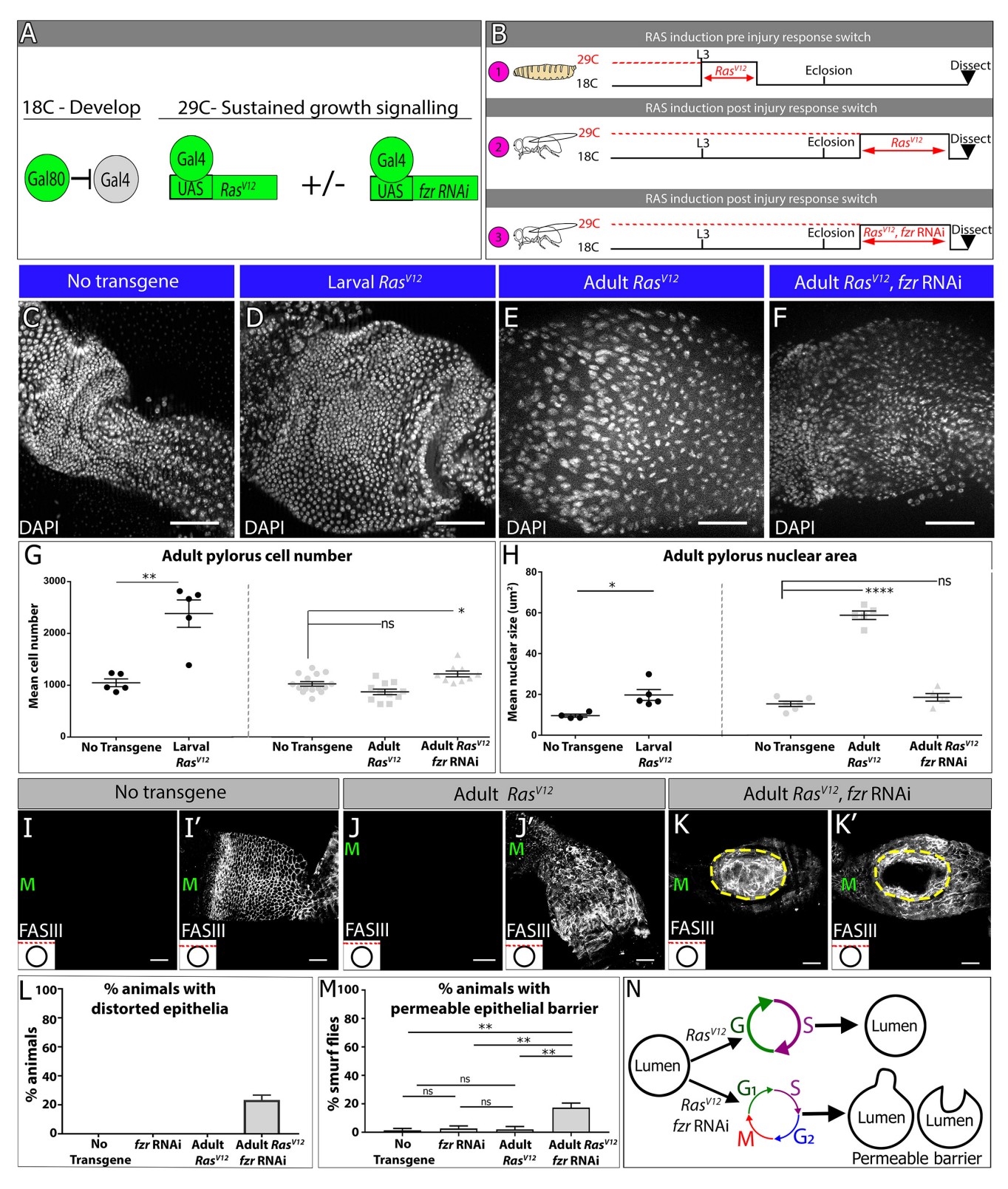

**Figure 5.** Endocycles preserve intestinal architecture in response to sustained tissue growth signaling. (**A**) Schematic and (**B**) experimental design of sustained growth signaling expression. Numbers 1–3 in (**B**) are referenced in the text. (**C–F**) Adult pyloric nuclei (DAPI, white) ±*Ras^{V12}* expression. (**C**) No transgene expression. (**D**) 48 hr *Ras^{V12}* expression at L3. (**E**) 6 days *Ras^{V12}* expression in adults. (**F**) 6 days *Ras^{V12}* expression in adults + *fzr RNAi*. (**G**–**H**) Quantification of adult pyloric cell number (**G**) and nuclear area (**H**). Data represent mean ± SEM. For adult induction, N ≥ 5 animals, at least two

*Figure 5 continued on next page*

*Figure 5 continued*

replicates. ANOVA with Dunnett's multiple comparisons; For larval induction, N ≥ 4 animals, at least two replicates. Unpaired two-tailed t-test. (I–K) Adult pylorus lateral membranes highlighted by FasIII staining (white). Distinct Z-sections of the hindgut positions shown in each image are illustrated by a red line in an inset in the bottom left, where a black circle represents the hindgut epithelium. 'M' indicates midgut position. (I–I') No transgene expressed. (J–J') 14 days $Ras^{V12}$ expression in adults. (K–K') 14 days $Ras^{V12}$ expression in adults + *fzr* RNAi. Yellow outline highlights distorted epithelia. (L) Quantification of % animals with epithelial distortions. Data represent mean ±SEM, N = 3 experiments, N ≥ 24 animals per condition. (M) Quantification of % animals with permeable epithelial barrier. Data represent mean ± SEM, N = 3 experiments, N ≥ 27 animals per condition. ANOVA with Tukey's multiple comparisons. (N) Model of tissue architecture in conditions of sustained growth signaling. Scale bars (C–F,I–K') 50 μm.
DOI: https://doi.org/10.7554/eLife.38327.012

The following figure supplement is available for figure 5:

**Figure supplement 1.** Supporting data related to *Figure 5*.
DOI: https://doi.org/10.7554/eLife.38327.013

---

increased tissue area in *fzr RNAi +RasV12* animals, as adult pyloric area actually decreases in these animals relative to animals expressing $Ras^{V12}$ alone (*Figure 5—figure supplement 1D*, see Discussion for proposed mechanism). One possible consequence of the altered intestinal epithelial architecture that we observe is a compromised epithelial barrier. We thus tested whether $Ras^{V12}$ and/or *fzr RNAi* expression alters intestinal barrier function. Using the well-established 'Smurf' assay (*Rera et al., 2011*; see Materials and methods) we observed that only *fzr RNAi +RasV12* animals show substantial permeability of the epithelial barrier. We note that the frequency of *fzr RNAi +-RasV12* animals with a permeable epithelial barrier (17.33% ± 3.18% SEM, *Figure 5M*) is very similar to the proportion of animals of this same genotype with epithelial architecture distortions (*Figure 5L* vs. M), implying that these defects may be related. Thus, under conditions of prolonged growth signaling, endocycles protect the pyloric epithelium against hyperplasia, distortions in epithelial lumen architecture, and gut barrier permeability (*Figure 5N*).

## Discussion

The diversity of cell cycle programming in mature tissues in response to tissue stresses such as injury or ectopic growth signaling are increasingly appreciated (*Øvrebø and Edgar, 2018*). Here, we use *Drosophila* pyloric cells as a model to understand not only the regulation, but also the implications, of using two different cell cycle programs in an adult tissue subject to injury/growth signaling stress. Capitalizing on this simplicity and our development of DEMISE as a highly manipulatable tissue injury system, here we delineated developmental and molecular parameters that determine whether *Drosophila* pyloric cells undergo mitotic cycles or ploidy-increasing endocycles. We then extended our findings by demonstrating that ploidy-increasing endocycles protect the adult pylorus, a normally quiescent tissue, from hyperplastic growth and epithelial architecture disruption. These findings highlight a potential protective effect of polyploidization over mitosis in a post-developmental setting, one that we suggest may prevent disease in diverse injured organs.

### Polyploidy as a recurring tissue injury response and tissue architecture preservation mechanism

To date, many studies have viewed ploidy increases following tissue injury as either maladaptive or neutral. In the vertebrate heart, ploidy increases cause organ hypertrophy and overgrowth (*González-Rosa et al., 2018*; *Li et al., 1996*; *Senyo et al., 2013*; *Steinhauser and Lee, 2011*), a disease phenotype that increases heart wall thickness while decreasing valve size (*Bonow et al., 2011*). Unlike the vertebrate heart, tissue overgrowth does not commonly occur in the injured mammalian liver, which undergoes differing degrees of hepatocyte ploidy increases upon injury. However, likely due to competing contributions from mitosis of diploid hepatocytes or from liver stem cells, as well as an inherent flexibility in the cell cycles used to regenerate the liver (*Chen et al., 2012*; *Diril et al., 2012*; *Lazzeri et al., 2018*; *Miyaoka et al., 2012*; *Nevzorova et al., 2009*; *Pandit et al., 2012*) a role for ploidy in the repairing liver remains unclear. Relative to other tissue models involving ploidy increases, the adult pyloric response is simpler, as it does not occur in parallel to cell-cell fusion (*Losick et al., 2013*; *Losick et al., 2016*), divisions of diploid stem cells (*Lin et al., 2018*; *Wang et al., 2015*) and does not occur in cells that are already programmed to become polyploid

regardless of injury (*Tamori and Deng, 2013*). Using our simpler system, we were able to more directly ask whether polyploidy has any advantage in tissue repair and tissue overgrowth, or rather is an aberrant response to injury.

Our work here suggests that polyploidy can represent a regulated and potential beneficial tissue injury response. First, using the simplicity of our system, we discovered that ploidy increases in the repairing pylorus are perfectly tuned to replace the pre-injury number of genomes. This result suggests that the number of post-injury endocycles is tightly regulated and can be responsive to injury severity. While diploid mitotic cycles replace pre-injury genomes in the larva, endocycles do the same in the adult. Further, reverting adult endocycles to mitotic cycles in the injured adult also leads to the correct number of pre-injury cells with no ploidy increase, suggesting that both mitotic cycles and endocycles are able to accurately replace lost tissue mass/genome content. Future work can determine whether endocycling is triggered by mechanical stress, as suggested by recent work in the repairing zebrafish epicardium (*Cao et al., 2017*). Additionally, it will be interesting to determine whether there is any advantage to skipping late S-phase replication during endocycles, which occurs in wild-type but not in *fzr* animals during repair. Our results also mirror findings in the injured *Drosophila* abdomen where ploidy matching was also observed. However, our analysis is not complicated by parallel cell-cell fusion events (*Losick et al., 2016*).

Second, using a $Ras^{V12}$ model, we present evidence that endocycles enable the pylorus to resist tissue malformation and permeability phenotypes under conditions of sustained tissue growth signaling. Our findings in the post-developmental pylorus may mimic the finding in developing glia of the *Drosophila* blood-brain barrier, in which endocycles preserve tissue integrity during growth (*Unhavaithaya and Orr-Weaver, 2012*; *Von Stetina et al., 2018*). As activated Ras/Raf signaling reproduces the injury responsive cell cycles in our system and in injured cardiac tissues, this model may suggest that endocycles are employed in some tissues to maintain epithelial integrity in the face of stresses such as injury or excess growth signaling. One form of tissue stress may come from cell shape changes during mitosis, which depend on regulated changes in cell adhesion (*Kunda et al., 2008*; *Lancaster et al., 2013*). Future work can determine if such mitotic shape changes are incompatible with preserving pyloric tissue architecture.

We note that the pylorus has a potentially important similarity to other tissues exhibiting injury-induced ploidy increases such as the mammalian heart, liver, and kidney: these tissues are all normally quiescent or exhibiting very low cell turnover. In such tissues, stem cell-based divisions are not quickly rejuvenating the cell population, and thus cells with de novo mutations (such as dominant *Ras* mutations) may accumulate during aging. As a result, these long-lived tissues may require a strategy to minimize mitosis of any cell. While such quiescent tissues are then less reliant on mitosis to restore tissue mass, they can still employ ploidy-increasing cycles to accomplish the same goal. In line with this idea, we note that of any organ, the heart has one of the lowest incidences of cancers (*Bisel et al., 1953*; *Leja et al., 2011*), and a recent study also showed a tumor-protective role of polyploidy in the liver (*Zhang et al., 2017*). Future work can determine whether tumor protection or preservation of epithelial integrity is a general property of tissues prone to injury- or growth signal-induced ploidy increases.

## Developmental vs. stress-induced endoreplication

Our results show that, while both injury-induced and developmental endocycles rely on Fzr, the upstream regulation in the injury context does not involve regulation of Notch, a well-known developmental endoreplication regulator in flies and mammals (*Cornejo et al., 2011*; *Deng et al., 2001*; *Domanitskaya and SchupbachSchüpbach, 2012*; *Mercher et al., 2008*; *Micchelli and Perrimon, 2006*; *Ohlstein and Spradling, 2006*; *Poirault-Chassac et al., 2010*; *Shcherbata et al., 2004*; *Sun and Deng, 2007*; *Von Stetina et al., 2018*). Future work in our system can determine what factors converge on *fzr* regulation and the control of entry into mitosis or endocycles following pyloric injury. One candidate is the ecdysone steroid hormone receptor, which peaks in activity during metamorphosis, close to when the pyloric cell cycle injury response changes. It is somewhat surprising that loss of *fzr* alone is sufficient to restore pyloric mitosis, as the cyclin dependent kinase 1 (Cdk1) activator Cdc25/String is often required (*Schaeffer et al., 2004*; *Von Stetina et al., 2018*). Thus, in the pylorus, Cdk1 may be primed for activity but is kept inactive without its cyclin binding partners, which are negatively regulated by high levels of APC/Fzr. More broadly, further study of

the mechanisms that alter cell cycle programming after tissue injury may also improve therapeutic efforts to regenerate injured organs.

Another question raised by our work is whether there are functional benefits to the altered pyloric cell cycle injury response. As our data suggest, in the normally quiescent adult stage, induced cell proliferation may be more detrimental for tissue integrity than induced endocycles. However, building new tissues during intestinal development necessitates the use of cell division. For poorly understood reasons, the Dipteran intestine is completely remodeled during metamorphosis, and cell divisions from the pylorus are the source of the majority of new hindgut cells (*Aghajanian et al., 2016*; *Bodenstein, 1950*; *Fox and Spradling, 2009*; *Robertson, 1936*; *Sawyer et al., 2017*; *Takashima et al., 2008*). This need for wholescale organ remodeling necessitates the ability to divide during larval/early pupal stages. Once organ remodeling is complete, the pylorus, which lacks stem cells, then ceases to divide (*Fox and Spradling, 2009*; *Sawyer et al., 2017*). Our data suggest that at this point, high *fzr* levels prevent any future cell division. Following injury, there may be an increased negative regulation of mitotic entry, as we and others have shown that polyploid cell division causes genomic instability (*Davoli et al., 2010*; *Duncan et al., 2010*; *Fox et al., 2010*; *Hassel et al., 2014*; *Schoenfelder et al., 2014*; *Storchová et al., 2006*). Future work can determine if mitotic integrity is compromised during mitotic re-entry in the adult pylorus.

## DEMISE- a versatile tool for tissue injury and cell death studies

In addition to the new tissue injury biology presented here, we also introduce DEMISE as a method for dual control over tissue injury and transgene expression. We show that this system is amenable to injury studies in the imaginal disc, a widely used tissue injury system (*Harris et al., 2016*; *Smith-Bolton et al., 2009*). Other tissues with injury responses that we have not tested but that would benefit from DEMISE as a tool include the stem cell-based midgut (*Apidianakis and Rahme, 2009*; *Buchon et al., 2009*; *Chatterjee and Ip, 2009*; *Jiang et al., 2009*), the brain (*Moreno et al., 2015*), the abdomen (*Losick et al., 2013*; *Losick et al., 2016*), the muscle (*Chaturvedi et al., 2017*) and ovarian follicle cells (*Tamori and Deng, 2013*). Beyond studies of tissue injury, our inducible *reaper* transgene can be used in studies of apoptotic signaling, such as the regulation of 'undead' cells which contain active caspase signaling but persist and influence the behavior of neighboring cells (*Deveraux et al., 1998*; *Fan and Bergmann, 2014*; *Hay et al., 1994*; *Ryoo et al., 2004*).

While many dual transgene systems in flies combine Gal4 with either LexA or Q systems (*Kockel et al., 2016*; *Lai and Lee, 2006*; *Potter et al., 2010*), there are vastly more currently available Gal4 and partner *UAS* lines than for these other systems, making our system more immediately employable for the most cell types/transgenes. Additionally, the use of a single Gal4 promoter for both injury and transgene activation provides tissue specificity and ensures both injury and transgenes are expressed in the same population of cells. However, use of the same promoter to drive injury and transgenes also creates a limitation to our system if one wants to study the response of cell population A on injury to population B. In such case, combining LexA/Q system injury with Gal4 transgene induction (or vice versa) would be necessary. Beyond *Drosophila*, we note that our system could be adapted for use in other organisms, by adapting existing FLP and Cre-mediated injury and gene knockout models.

In summary, this study highlights the utility of the *Drosophila* pylorus in identifying regulation and purposes of specific cell cycle programs induced by tissue injury. Future work in this system can continue to illuminate the role and regulation of variant cell cycles and polyploidy in tissue injury biology.

## Materials and methods

**Key resources table**

| Reagent type (species) | Designation | Source or reference | Identifiers | Additional information |
|---|---|---|---|---|
| gene (D. melanogaster) | *fzr* | NA | FBgn0262699 | |
| genetic reagent (D. melanogaster) | *byn > Gal4* | Singer et al.,1996 | FBal0137290 | P{GawB}bynGal4 |

*Continued on next page*

*Continued*

| Reagent type (species) | Designation | Source or reference | Identifiers | Additional information |
|---|---|---|---|---|
| genetic reagent (D. melanogaster) | act >> LacZ | BDSC | Stock 6355 | P{ry[+t7.2]=Act5C (FRT.polyA)lacZ.nls1}3, ry[506] |
| genetic reagent (D. melanogaster) | hsFLP12;Sco/CyO | BDSC | Stock 1929 | P{ry[+t7.2]=hsFLP} 12, y[1] w[*]; sna[Sco]/CyO |
| genetic reagent (D. melanogaster) | UAS fzr RNAi | VDRC | Stock 25550 | w1118; P{GD9960}v25550 |
| genetic reagent (D. melanogaster) | UAS-Ras85D$^{V12}$ | BDSC | Stock 4847 | w[1118]; P{w[+mC]=UAS-Ras85D.V12}TL1 |
| genetic reagent (D. melanogaster) | UAS-hid, UAS-reaper | Zhou et al 1997 | | |
| antibody | anti-Fasciclin III | DSHB | 7G10 | |
| antibody | anti-Phospho-Histone H3 | Cell Signaling | ab9706 | |
| antibody | anti-BrdU | Serotec | 3J9 | |
| antibody | anti-Beta-Galactosidase | Abcam | ab9361 | |
| recombinant DNA reagent | pUAST-FRT-Stop-FRT-rpr (DEMISE) plasmid | this paper | | progenitor: pUAST-FRT-Stop-FRT-mCD8-GFP |
| recombinant DNA reagent | pUAST-FRT-Stop-FRT-mCD8-GFP plasmid | Potter et al., 2010 | addgene #24385 | |
| commercial assay or kit | TUNEL | Roche | #12156792910 | |

## Fly stocks

Full genotypes are described at flybase.org. Except where indicated, flies were raised at 25C on standard *Drosophila* media (Archon Scientific, Durham). For larval experiments, animals were collected at wandering 3rd instar stage (L3). All adults dissected were older than 4 days. The following publicly available stocks were used in the study: *wg > LacZ* (#BS 1567), *act >> LacZ*(#BS 6355), *hsFLP12;Sco/ CyO* (#BS 1929), *UAS fzr RNAi* (#VDRC 25550), *fzr$^{G0418}$* (#BS12297), *fzr$^{G0326}$* (BS#12241) and *UAS-Ras85D$^{V12}$* (#BS) where BS = Bloomington Stock Center and VDRC = Vienna Drosophila Resource Center. The following fly stocks were generously gifted to us: *byn >Gal4* (Shigeo Takashima- UCLA, Singer et al., 1996), *fz3RFP* (Andrea Page-McCaw, Vanderbilt), *P(Vha16-$^{1CA06708}$)/CyO* (Carnegie Protein Trap Collection), *UAS-hid, UAS-reaper* (Zhou et al., 1997, Toshie Kai, Temasek Laboratory), *UAS N RNAi* and *UAS N-DN* (Sarah Bray, University of Cambridge).

## *Drosophila* genetics

All *UAS* transgenes were induced by *byn > Gal4* or *en > Gal4.* All temporally controlled *UAS*-transgene experiments involved culturing flies with a Gal80$^{ts}$ repressor driven by the tubulin promoter in the genetic background at 18C except during the desired period of expression, during which they were transferred to 29C as described in the appropriate figure panel and accompanying text. For adult injury, after eclosion, all animals were aged 4–7 days at 18C before injury as done previously (Fox and Spradling, 2009; Losick et al., 2013; Sawyer et al., 2017). For larval injury, we shifted animals at L3 stage to 29C for 16 hr, a sub lethal level of injury induction. Heat shock conditions to induce FLP were performed at 37C by heat shocking either three times for 45 min each with a 1–2 hr recovery in between (DEMISE experiments) or a single time for 25 min (cell lineage tracing experiments).

## DEMISE

The DEMISE plasmid was generated by restriction digest of *pUAST > Stop > mCD8-GFP* plasmid (addgene #24385, Potter et al., 2010) using Xhol and Stul. An insert containing a *rpr*-cDNA, SV40NLS and SV40PolyA sequences was then synthesized to make a final vector *pUAST-FRT-Stop-FRT-rpr* (Bio Basic Inc, New York). The construct was then injected to embryos and six fly lines were selected to include the construct on either the 2nd or 3rd chromosome. DEMISE flies (*pUAST-FRT-Stop-FRT-rpr/CyO; byn >Gal4,Tub >Gal80/TM6*) were raised at 25C unless otherwise noted. Flies

were crossed to *hsFLP[12]* to induce cell death. Apoptotic death was observed in the wing disc in less than 48 hr, while in the hindgut it was observed in less than 24 hr.

## Antibodies and cell markers

Markers for ileal cells (*Vha16[Ca06708]*) and the hindgut-midgut boundary (*wg-LacZ or fz3-RFP*) were used to specify the pylorus from adjacent regions of the gut. BrdU labeling and colcemid (Sigma) feeding were performed as in *Sawyer et al. (2017)*. For cell death quantifications, TUNEL was performed using in situ cell death detection kit (Roche, Basel, Switzerland) as described previously (*Schoenfelder et al., 2014*). For antibody staining, tissues were dissected in 1X PBS and immediately fixed in 1XPBS, 3.7% paraformaldehyde, 0.3% Triton-X for 30–45 min. Immunostaining was performed as described in *Sawyer et al. (2017)*. The following antibodies were used in this study: Fasciclin III (FasIII, DSHB, 7G10, 1:50), Beta-Galactosidase (Abcam, ab9361, 1:1000), DCP1 (Cell Signaling, Asp261, 1:1000), BrdU (Serotec, 3J9, 1:200), Phospho-Histone H3 (Cell Signaling, #9706, 1:1000), Centrosomin (generous gift from Nasser Rusan lab, NIH/NHLBI, 1:10,000). All secondary antibodies used were Alexa Fluor dyes (Invitrogen, 1:500). Tissues were mounted in Vectashield (Vector Laboratories Inc.). Images were acquired with the following: an upright Zeiss AxioImager M.2 with Apotome processing (10X NA 0.3 EC Plan-Neofluar air lens or 20X NA 0.5 EC Plan-Neofluar air lens) or inverted Leica SP5 (40X NA 1.25 HCX PL APO oil immersion). Image analysis was performed using ImageJ (*Schneider et al., 2012*), including adjusting brightness/contract, Z projections, cell counts, cell area and integrated density quantification. DNA FISH was performed as in *Beliveau et al. (2014)*. Cy5-labeled oligo-probes to the AACAC repeat were synthesized by Integrated DNA Technologies.

## Ploidy measurements

For ploidy measurements, guts were dissected in 1X PBS and prepared as described previously (*Losick et al., 2013*). Total tissue ploidy was calculated by timing the average animal ploidy per injury condition with the average recovered cell number $\pm$ STERR (DNA content * cell count).

## Smurf assay for epithelial barrier integrity measurements

For measuring epithelial barrier integrity, we adapted the established Smurf assay (*Rera et al., 2011*). Flies were raised at 18C on standard *Drosophila* media (Archon Scientific, Durham) until 4–7 days post eclosion. Flies were then shifted to vials containing standard *Drosophila* media mixed with 0.5% Bromophenol blue, and were then kept at 29C and scored for gut permeability every 24 hr for 14 days.

## Cell cycle nomenclature

(C)refers to the haploid DNA content. We define a 'polyploid' cell as a somatic cell that contains more than the diploid number of chromosome sets. We define the 'endocycle' as any programmed cell cycle in which the genome reduplicates without any sign of mitotic entry. We define 'endoreplication' as a broader term encompassing any truncated cycle that generates polyploid cells. We note the use of other terms in the literature that refer to similar processes, and also note that such terms are not always used consistently. We thus adopt the terms used most consistently in the current literature (*Edgar et al., 2014*).

## Statistical analysis and reporting

Statistical analysis was performed using GraphPad Prism 7. Statistical tests and adjustments of P-values for multiple comparisons are detailed in figure legends. For all tests, P value reporting is as follows: ($p>0.05$,ns); ($p<0.05$,*); ($p<0.01$,**); ($p<0.001$,***); ($p<0.0001$, ****). Regression analysis for ploidy measurements and cell number was done using the formula log2(ploidy)x cell number, and a Pearson correlation analysis.

## Acknowlegements

We thank Jumana Abed, Nasser Rusan, and Ting Wu for providing useful reagents and technical advice. We thank Bernard Mathey-Prevot, David MacAlpine, Jingli Cao, and Fox laboratory members for comments on the manuscript. This project was supported by NIGMS grant GM118447 to DF.

## Additional information

### Funding

| Funder | Grant reference number | Author |
|---|---|---|
| National Institutes of Health | GM118447 | Donald T Fox |

The funders had no role in study design, data collection and interpretation, or the decision to submit the work for publication.

### Author contributions

Erez Cohen, Conceptualization, Resources, Data curation, Software, Formal analysis, Supervision, Validation, Investigation, Visualization, Methodology, Writing—original draft, Project administration, Writing—review and editing; Scott R Allen, Conceptualization, Data curation, Formal analysis, Investigation, Visualization, Methodology; Jessica K Sawyer, Conceptualization, Formal analysis, Investigation, Visualization; Donald T Fox, Conceptualization, Resources, Formal analysis, Supervision, Funding acquisition, Validation, Investigation, Visualization, Writing—original draft, Project administration, Writing—review and editing

### Author ORCIDs

Erez Cohen (iD) http://orcid.org/0000-0003-2390-3707
Scott R Allen (iD) http://orcid.org/0000-0002-4809-0493
Donald T Fox (iD) http://orcid.org/0000-0002-0436-179X

### Decision letter and Author response

Decision letter https://doi.org/10.7554/eLife.38327.016
Author response https://doi.org/10.7554/eLife.38327.017

## Additional files

### Supplementary files

• Transparent reporting form
DOI: https://doi.org/10.7554/eLife.38327.014

### Data availability

All data generated or analysed during this study are included in the manuscript and supporting files.

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
