## [Decision Letter]

Thank you for submitting your article "A Cell Cycle Switch Dictates Organ Repair and Tissue Growth Responses in the *Drosophila* Hindgut" for consideration by *eLife*. Your article has been reviewed by Marianne Bronner as the Senior Editor, a Reviewing Editor, and three reviewers. The following individual involved in review of your submission has agreed to reveal his identity: Norman Zielke (Reviewer #1).

The reviewers have discussed the reviews with one another and the Reviewing Editor has drafted this decision to help you prepare a revised submission.

Summary:

This manuscript details the responses to injury of the larval and adult pyloris of *Drosophila*, and highlights two different modes of tissue repair, one employing mitotic cell proliferation (in the larva) and the other employing endocycling (in the adult). The authors provide ample, high quality data showing that this cell cycle switch is controlled mediated by Fizzy-related (Fzr), a known regulator of mitotic to endocycle transitions. The three reviewers were uniformly enthusiastic about the quality and significance of the manuscript and suggested relatively minor clarifications be added in revision.

Essential revisions:

1) The revisions suggested in the reviews, appended below in full, are nearly all editorial and should be simple to address in revision. Hopefully you can address all of the clarifications and corrections not requiring new data in your revision.

As reviewing editor, I also suggest that you try the gut permeability experiment requested by reviewer 3. This should be straightforward as protocols for such assays (the "smurf" assay) already exist. However, if this experiment proves technically difficult, too time consuming, or uninformative, it will not be required for publication.

2) In addition, I'd like to mention one point that the reviews did not address, namely how Fzr is controlled developmentally. Is the switch to a Fzr-regulated endocycle mediated by induction of Fzr sometime during metamorphosis? Although I did see that you rule out hindsight and Notch as regulators, and propose a potential role for Ecdysone and EcR, I did not find data on Fzr protein or RNA expression in the pyloris during the transition. I believe such data are important for this story and should be included in some form. In your revision, please describe the pattern of Fzr expression by including new data, or the relevant citations if they exist. If these data are already included, I apologize for the oversight.

Reviewer #1

The manuscript "A Cell Cycle Switch Dictates Organ Repair and Tissue Growth Responses in The *Drosophila* Hindgut" by Cohen et al., describes the cell cycle response to acute cell ablation in the *Drosophila* hindgut. The manuscript is well written and the included data are of very high quality and accurately reflects the conclusions in the text. However, most of the obtained results are somewhat expected from the data that have been obtained in other model systems and/or context. The most striking finding of this study is the observation that tissue regeneration upon acute cell ablation in the adult pylorus of Fzr-depleted animals is mediated by mitotic cycles although these cells are normally primed to regenerate via endocycles. This is a significant finding, and thus this manuscript should qualify for publication in *eLife* after the minor problems listed below have been addressed.

- To manipulate the tissue in a precise manner the authors have developed a new Flp/FRT-Gal80^ts^-based method termed DEMISE, which enables spatiotemporally controlled induction of the pro-apoptotic gene reaper as well as independent expression of UAS-transgenes such as Fzr-RNAi. This is a very elegant approach, but in my opinion does acute cell ablation with reaper not exactly reflect the kind of damage to which the cells in the hindgut are normally exposed? A hallmark of ageing is loss of intestinal integrity due to accumulation of cells with damaged DNA. Earlier work in hepatocytes with dysfunctional telomeres (Lazzerini-Denchi et al., 2006) demonstrated that liver regeneration in presence of DNA damage signals is mediated by endocycles, which led to the notion that damaged, but viable cells utilize endocycles instead of mitotic cycles to avoid mitotic errors and propagation of the damaged DNA. Therefore, the finding that acute cell ablation in the larval pylorus is compensated by additional mitotic cycles, could be rather a normal response to cell ablation in developing tissues, and thus not per se exclude that these cells could regenerate through endocycles. Therefore, it would be advisable to replace reaper by Cas9 and a "sloppy"-gRNA that introduces DSBs throughout the genome, to re-evaluate (perhaps in a further study) the question whether the cells in the larval pylorus have the capacity to regenerate by endocycles.

- In subsection “DEMISE reveals *fizzy-related* as a regulator of injury-mediated cell cycle programming”, the authors state "Our pre-injury FUCCI data (Figure 2I-K) provided a potential clue, as larval but not adult uninjured pyloric cells express the mitotic regulator CyclinB", which is misleading as the Fly-FUCCI system doesn't reflect the expression levels of endogenous CyclinB. To make this statement the authors should utilize a Cyclin B antibody or a GFP-Trap.

- In subsection “Developmental vs. Stress-induced endoreplication”, the authors state "It is somewhat surprising that loss of *fzr* alone is sufficient to restore pyloric mitosis, as the cyclin dependent kinase 1 (Cdk1) activator Cdc25/String is often required (Schaeffer et al., 2004; Von Stetina et al., 2018)". Stg/Cdc25 is an APC/C-Fzr target in the *Drosophila* embryo (Reber et al., 2006), so provided the Stg/Cdc25 transcript is present in these cells, Fzr-depletion could potentially lead also to accumulation Stg/Cdc25 and thereby promote the G2/M transition. Therefore, the authors should evaluate Stg/Cdc25 levels by antibody staining or GFP-trap.

Reviewer #2

A variety of distinct cell cycle types apart from the canonical mitotic cell division cycle are encountered in eukaryotes. The manuscript by Cohen et al. reports experimental work addressing regulation and purpose of a switch from canonical to endocycle. The same lab has previously published a series of papers on this and related issues, primarily focusing on the hindgut in *Drosophila* larvae and adults. The current study builds on their steadily accumulated outstanding expertise. Their previous work has shown that pylorus cells in the adult hindgut respond to experimentally induced apoptosis that kills a substantial fraction of these cells. The surviving cells repair and restore the lost tissue mass by re-growing while progressing through endocycles. Here Cohen et al. compare the response to analogous experimentally induced apoptosis in the larval and adult pylorus. Well controlled experiments (expression of pro-apoptotic proteins in a tissue-specific and temporally controlled manner with the help of GAL4/GAL80^ts^) resulting in highly convincing data demonstrate clearly that tissue mass and in particular total gene content (DNA amount) is restored in both cases but via different routes: cell proliferation (growth paralleled by progression through mitotic cell cycles) in larvae versus endocycles (growth paralleled by genome amplification in the absence of mitosis) in adults.

In the second part of the manuscript the authors switch to a more sophisticated control of induced apoptosis. By combining GAL4/GAL80^ts^ with Flp/FRT regulation they achieve experimental induction of apoptosis in a few random cells within a tissue where additional transgenes (including UAS-RNAi for knock down of some gene of interest) can be expressed (with tissue-specific and temporal control). They designate their control system as DEMISE. They exploit DEMISE for an analysis of the role of a particular cell cycle regulator that has been most consistently implicated in mitotic->endocycle switching in many labs and experimental systems. They arrive at a convincing demonstration that knockdown of this APC/C regulator Fzr/Cdh1 in the adult pylorus converts the repair route from adult to larval style.

Cohen et al., do not report about efforts towards clarification whether this change in the repair mode in the adult pylorus has any physiological consequences on the function of the repaired organ. Instead they switch to another type of damage in the final part of the manuscript. Instead of expressing a pro-apoptotic protein they now express activated Ras (Ras^V12^) in larval or adult pylorus cells. This is shown to result in excessive growth accompanied by progression through mitotic cycles in larvae and through endocycles in adults. Again Fzr/Cdh1 knockdown is shown to convert these endocycles into mitotic cycles in adults. Finally, it is shown that repair of adult pylorus via mitotic cycles results in morphological organ deformations that are not observed after repair via endocycles, suggesting that an increased risk of tissue destabilization by mitosis/cytokinesis might explain why evolution might have favored repair by endocycles. While this proposal is not really new, I am not aware of any supportive evidence that comes anywhere close to what is nicely demonstrated here. While it can be debated whether the experimentally induced damage has much to do with physiological insults, it has certainly allowed precise and conclusive analyses. The writing is equally precise and convincing.

Reviewer #3

The manuscript by Cohen et al., reports that the *Drosophila* pylorus has different types of regeneration after injury that depends on developmental stage. The larval pylorus regenerates by mitosis, whereas the adult pylorus regenerates by endoreplication cell cycles. They show that the ubiquitin ligase Fzr (Cdh1), but not Notch, promotes endoreplication in the adult. Knockdown of Fzr caused adult pylorus cells to regenerate tissue mass by mitotic cell cycles instead of endoreplication cycles. They also showed that endoreplication cycles, but not mitotic cycles, in the adult pylorus resist perturbation of epithelial architecture caused by promotion of Ras^V12^ over-expression. Along the way they develop a new system they call DEMISE, which permits independent induction of transgene expression and cell death to evaluate the contribution of genes and growth programs to tissue regeneration.

This is a quality study that reports very interesting findings relevant to the question of how the response to tissue injury differs during development. It should be of interest to a fairly broad readership. The experiments are well controlled, the manuscript is well-written, and it contains provocative ideas, but the discussion is a tad on the long side. I suggest below one simple experiment and have a few suggestions and questions.

1) A major finding is that adult pyloric epithelial architecture is altered after Fzr RNAi and Ras^V12^ overexpression. The authors speculate that this may compromise epithelial integrity. The authors should assess whether the hyperplasia increases pyloric epithelial permeability using standard dye or other assays.

2) Is the epithelial outpouching/inpouching in Ras^V12^ Fzr RNAi animals associated with piling up of epithelial cells?

3) Was the combined effect of Ras^V12^ over-expression and cell death tested, +/- Fzr RNAi, using the new DEMISE injury system?

4) The authors discuss that polyploid cycles in the adult "skip" late replication. Please be clear whether this means a change in replication timing of genomic regions or under-replication as in other polyploid cells. It seems that the EdU labeling does not distinguish between these possibilities.

5) The authors define their cell cycle nomenclature in the methods: "We define the "endocycle" as any programmed cell cycle in which the genome reduplicates but chromosomes are not segregated into two daughter cells." That seems like too broad of a definition that would include cells that pass through various stages of an abortive mitosis / cytokinesis but fail to divide. These types of cycles are often called endomitotic. I know that there is not unanimity on the use of this terminology in the field, but are the authors promoting that all polyploid cycles with periodic S phase be called endocycles?

---

## [Author Response]

[…] Essential revisions:1) The revisions suggested in the reviews, appended below in full, are nearly all editorial and should be simple to address in revision. Hopefully you can address all of the clarifications and corrections not requiring new data in your revision.As reviewing editor, I also suggest that you try the gut permeability experiment requested by reviewer 3. This should be straightforward as protocols for such assays (the "smurf" assay) already exist. However, if this experiment proves technically difficult, too time consuming, or uninformative, it will not be required for publication.

As requested, we now include the experiment suggested by reviewer 3. In agreement with our finding that ~20% of *Ras^V12^ + fzr RNAi* animals have perturbed architecture, we now show that ~20% of *Ras^V12^ + fzr RNAi* animals also have lost gut permeability, whereas we observed no substantial permeability phenotypes with either *Ras^V12^* or control animals. We include these new data in Figure 5M. (Previous Figure 5M changed to Figure 5L).

2) In addition, I'd like to mention one point that the reviews did not address, namely how Fzr is controlled developmentally. Is the switch to a Fzr-regulated endocycle mediated by induction of Fzr sometime during metamorphosis? Although I did see that you rule out hindsight and Notch as regulators, and propose a potential role for ecdysone and EcR, I did not find data on Fzr protein or RNA expression in the pyloris during the transition. I believe such data are important for this story and should be included in some form. In your revision, please describe the pattern of Fzr expression by including new data, or the relevant citations if they exist. If these data are already included, I apologize for the oversight.

As requested, we now include data on Fzr expression in the pylorus. Using two previously established *fzr* transcriptional reporters, we find that *fzr* levels are undetectable in the larval pylorus, but are detectable in adjacent endocycling cells. In contrast, *fzr* is easily detectable in adult pyloric cells. These data mirror our functional data, which suggested that *fzr* is up-regulated in adult pyloric cells. We include these new data in Figure 4 and Figure 4—figure supplement 1.

Reviewer #1:[…] To manipulate the tissue in a precise manner the authors have developed a new Flp/FRT-Gal80^ts^-based method termed DEMISE, which enables spatiotemporally controlled induction of the pro-apoptotic gene reaper as well as independent expression of UAS-transgenes such as Fzr-RNAi. This is a very elegant approach, but in my opinion does acute cell ablation with reaper not exactly reflect the kind of damage to which the cells in the hindgut are normally exposed? A hallmark of ageing is loss of intestinal integrity due to accumulation of cells with damaged DNA. Earlier work in hepatocytes with dysfunctional telomeres (Lazzerini-Denchi et al., 2006) demonstrated that liver regeneration in presence of DNA damage signals is mediated by endocycles, which led to the notion that damaged, but viable cells utilize endocycles instead of mitotic cycles to avoid mitotic errors and propagation of the damaged DNA. Therefore, the finding that acute cell ablation in the larval pylorus is compensated by additional mitotic cycles, could be rather a normal response to cell ablation in developing tissues, and thus not per se exclude that these cells could regenerate through endocycles. Therefore, it would be advisable to replace reaper by Cas9 and a "sloppy"-gRNA that introduces DSBs throughout the genome, to re-evaluate (perhaps in a further study) the question whether the cells in the larval pylorus have the capacity to regenerate by endocycles.

Thank you for this suggestion, which we agree would be best suited for a further study. However, we note that as part of our previous study of DSB responses in the hindgut (Bretscher and Fox, 2016), we noted that the pylorus of larvae subjected to high levels (20 Gy) of X-irradiation, which also induces DSBs throughout the genome, appears to remain diploid. We noted the same result for animals which expressed the endonuclease ICre, which creates DSBs throughout the rDNA. These data indirectly support the idea that DSBs do not induce endocycles in the larval pylorus. Future in-depth studies can clarify whether the DSB response in the larval pylorus mirrors the apoptotic response that we report here.

- In subsection “DEMISE reveals fizzy-related as a regulator of injury-mediated cell cycle programming”, the authors state "Our pre-injury FUCCI data (Figure 2I-K) provided a potential clue, as larval but not adult uninjured pyloric cells express the mitotic regulator CyclinB", which is misleading as the Fly-FUCCI system doesn't reflect the expression levels of endogenous CyclinB. To make this statement the authors should utilize a Cyclin B antibody or a GFP-Trap.

Thank you for pointing this out. We were unable to detect CyclinB protein in the hindgut with either an available DSHB CyclinB antibody or in animals expressing Ubi-CyclinB GFP. These data may suggest that CyclinB levels in the hindgut are lower than in other tissues, or that technical challenges prevented us from visualizing CyclinB in this tissue. In line with this reviewer’s comment #5 (see below), we have resolved this issue by revising all references to the cyclinB-based FUCCI transgene in the text to say “Cyclin B_1-266_-mRFP.”

- In subsection “Developmental vs. Stress-induced endoreplication”, the authors state "It is somewhat surprising that loss of fzr alone is sufficient to restore pyloric mitosis, as the cyclin dependent kinase 1 (Cdk1) activator Cdc25/String is often required (Schaeffer et al., 2004; Von Stetina et al., 2018)". Stg/Cdc25 is an APC/C-Fzr target in the Drosophila embryo (Reber et al., 2006), so provided the Stg/Cdc25 transcript is present in these cells, Fzr-depletion could potentially lead also to accumulation Stg/Cdc25 and thereby promote the G2/M transition. Therefore, the authors should evaluate Stg/Cdc25 levels by antibody staining or GFP-trap.

As requested, we performed the reviewer’s suggested experiment. Much like the CyclinB experiment, we were unable to detect Stg protein in the hindgut of *fzr* animals (using an antibody to Stg generated by our colleague Stefano Di Talia). However, we were also unable to detect Stg in ANY tissue outside of the embryo. We also obtained similar negative results in the hindgut with available *stg* transgenic reporters, and thus we are hesitant to make any conclusions about Stg levels in the hindgut at this time.

Reviewer #3[…] This is a quality study that reports very interesting findings relevant to the question of how the response to tissue injury differs during development. It should be of interest to a fairly broad readership. The experiments are well controlled, the manuscript is well-written, and it contains provocative ideas, but the discussion is a tad on the long side. I suggest below one simple experiment and have a few suggestions and questions.1) A major finding is that adult pyloric epithelial architecture is altered after Fzr RNAi and Ras^V12^ overexpression. The authors speculate that this may compromise epithelial integrity. The authors should assess whether the hyperplasia increases pyloric epithelial permeability using standard dye or other assays.

Please see the response to “essential revisions”, #1.

2) Is the epithelial outpouching / inpouching in Ras^V12^ Fzr RNAi animals associated with piling up of epithelial cells?

Thank you for this question. We hope to investigate this question in depth in the future using light sheet microscopy.

3) Was the combined effect of Ras^V12^ over-expression and cell death tested, +/- Fzr RNAi, using the new DEMISE injury system?

We attempted to construct these 6-transgene animals prior to our first manuscript submission. However, in multiple attempts to construct these flies, we were unable to obtain viable animals needed to do this experiment. In parallel, we also attempted to synthesize this genotype in a clonal pattern using FLP/FRT but were also unsuccessful due to transgene leakage issues.

4) The authors discuss that polyploid cycles in the adult "skip" late replication. Please be clear whether this means a change in replication timing of genomic regions or under-replication as in other polyploid cells. It seems that the EdU labeling does not distinguish between these possibilities.

As requested, we now include new data using DNA FISH, which supports our prior claim that adult pyloric cells skip late replication during injury-induced endocycles. These new data are included in Figure 4—figure supplement 1.

5) The authors define their cell cycle nomenclature in the methods: "We define the "endocycle" as any programmed cell cycle in which the genome reduplicates but chromosomes are not segregated into two daughter cells." That seems like too broad of a definition that would include cells that pass through various stages of an abortive mitosis / cytokinesis but fail to divide. These types of cycles are often called endomitotic. I know that there is not unanimity on the use of this terminology in the field, but are the authors promoting that all polyploid cycles with periodic S phase be called endocycles?

As requested, we revised our endocycle definition to say “any programmed cell cycle in which the genome reduplicates without any sign of mitotic entry.”